# HUSP: A Smart Haptic Probe for Reliable Training in Musculoskeletal Evaluation Using Motion Sensors

**DOI:** 10.3390/s19010101

**Published:** 2018-12-29

**Authors:** Vidal Moreno, Belén Curto, Juan A. Garcia-Esteban, Felipe Hernández Zaballos, Pablo Alonso Hernández, F. Javier Serrano

**Affiliations:** Department of Computer Science and Automation, University of Salamanca, 37008 Salamanca, Spain; vmoreno@usal.es (V.M.); bcurto@usal.es (B.C.); jage@usal.es (J.A.G.-E.); felipezaballos@gmail.com (F.H.Z.); pablo.alonso.hernandez@gmail.com (P.A.H.)

**Keywords:** IMU sensors, haptic devices, USI based diagnostic

## Abstract

As a consequence of the huge development of IMU (Inertial Measurement Unit) sensors based on MEMS (Micro-Electromechanical Systems), innovative applications related to the analysis of human motion are now possible. In this paper, we present one of these applications: a portable platform for training in Ultrasound Imaging-based musculoskeletal (MSK) exploration in rehabilitation settings. Ultrasound Imaging (USI) in the diagnostic and treatment of MSK pathologies offers various advantages, but it is a strongly operator-dependent technique, so training and experience become of fundamental relevance for rehabilitation specialists. The key element of our platform is a replica of a real transducer (HUSP—Haptic US Probe), equipped with MEMS based IMU sensors, an embedded computing board to calculate its 3D orientation and a mouse board to obtain its relative position in the 2D plane. The sensor fusion algorithm used to resolve in real-time the 3D orientation (roll, pitch and yaw angles) of the probe from accelerometer, gyroscope and magnetometer data will be presented. Thanks to the results obtained, the integration of the probe into the learning platform allows a haptic sensation to be recreated in the rehabilitation trainee, with an attractive performance/cost ratio.

## 1. Introduction

Throughout our lives, practically everyone will experience an incident that will affect our musculoskeletal (MSK) or neuromuscular system. At these issues, the effects can take the form of acute injuries, such as fractures, breaks or muscle tears, sprains, tendinitis, cerebrovascular accidents with peripheral muscular paralysis, etc. In addition, they can evolve as subacute or chronic processes such as degenerative processes, rheumatological diseases or neuropathies with muscular involvement.

The impact of these diseases on our functional ability depends not only on the severity of the process but also on the treatment that will be applied. In this way, rehabilitation treatments play a fundamental role in the results that follow any type of orthopedic or sports injury or perhaps a surgical procedure. In these cases, several basic goals must be pursued, such as early mobilization and the establishment of adequate objectives to carry out such mobilization [1]. Many people with MSK pathologies benefit from the association of rehabilitation along with complementary medical treatments, so symptoms and disability are reduced, improving functional ability [2].

Prior to the establishment of a treatment, both medical or rehabilitative, a correct diagnosis is essential. The diagnostic task begins with an appropriate medical history, which is, usually, complemented with various imaging tests. These ones include simple X-ray, axial tomography, magnetic resonance or USI (UltraSound Imaging). USI is a non-invasive, safe (there is no radiation), repeatable technique and provides a dynamic study; therefore, it is widely accepted by both patients and medical services. In addition, thanks to technological advances, the resolution of USI equipment has been improved remarkably, hence it provides better quality images.

In the last decade [3,4], the use of ultrasound for imaging in the context of MSK evaluation in rehabilitation environments has expanded dramatically because it allows high-resolution images to be available in real time. Pioneering applications of ultrasound at MSK studies date back from 1970, for research on the rotator cuff [5]. Nowdays, this image technology adds accuracy on diagnosis and certainty to guide the needle insertion in therapeutic interventional procedures [6] such as drainage, infiltrations, etc. These benefits have led, in advanced countries, to its growing use in MSK clinics and rehabilitation installations, by providing clinical, anatomical, and technical integration in a one-day-evaluation [6].

Despite its obvious benefits, USI is a strongly operator-dependent technique, both at the image capturing stage and at the following evaluation stage. The sonoanatomy of MSK structures differs from magnetic resonance and radiographic and tomographic images, in such a way that USI images are harder to understand [6]. Firstly, relating 2D ultrasound imaging to a familiar 3D anatomical model is not a trivial issue. This difficulty can only be overcome by training, hence experience becomes of fundamental relevance [7]. In fact, USI technique is being incorporated into numerous university education curricula in different medical specialties, as it has proven to be very useful along with physical exploration [8]. The physician must learn to properly establish the position and orientation of the transducer (probe), and then he has to be able to interpret the obtained images. To do this task, it is necessary to get a correct mental model, that is, the relationship between hand movements and probe orientation with the resulting US image, that will let the learning curve improve [9]. The cost of equipment is high enough to make it impossible to have a device available for individual practice for medicine students. At present, the best way to acquire a good mental model in USI is a procedure with a simulator, which can transfer the knowledge from an expert [10]. At this procedure, haptics-based simulators provide a realistic feedback mechanism that allows the trainees to learn from their mistakes and to improve their technique, accelerating their learning process to develop accurate skills, as at our case of study, on MSK diagnosis.

The aim of this paper is to present a tool to train or instruct beginners in the recognition of different MSK tissues through the visualization of a US image. Our solution consists of two components: a probe replica that mimics a real US transducer and a multi-platform desktop application. The probe replica, referred to as HUSP (Haptic US Probe), will be equipped with sensors to capture the movement of a trainee’s hand. The software application reproduces US images, pre-recorded with real equipment, according to the probe movements as it is manipulated by the trainee. The portable nature of the device HUSP facilitates in-home training for the diagnosis of MSK diseases outside clinical and academical institutions. Obviously, the system applicability is not restricted to this medical diagnostic activity area.

Our innovative application needs to estimate the orientation (roll, pitch, yaw) of the HUSP. Inertial sensors can be used to capture data about the orientation of the object that includes the sensor. As a consequence of the huge development in Micro-electromechanical Systems (MEMS) technologies, IMU (Inertial Measurement Unit) sensors become more accurate, lightweight, compact and inexpensive. In [11], a complete review of the pose estimation methods using MEMS devices can be found.

The measurements of the gyroscope, magnetometer and accelerometer present errors due to bias, noise, magnetic interferences and other external disturbances [11]. Hence, their accuracy of orientation estimation may be poor if these measurements are considered individually. To improve the orientation estimation accuracy, sensor fusion algorithms (SFAs) are necessary, where accelerometer and magnetometer measurements are used for compensating the drift during gyroscope data integration. In [12], a comprehensive and systematic review of existing methods to reduce the effect of indoor magnetic disturbances is presented. Following this systematic review, we are going to categorize the SFA proposed in our work to estimate the 3D orientation of the probe.

Among the existing methods to decouple the estimation of the pitch and roll angles from the magnetic disturbance, our method uses the two-step orientation estimation. First, the pitch and roll angles using acceleration and angular velocity are estimated, and then yaw angle with the estimated pitch, roll angles and the magnetometer data are calculated. Ref. [13] proposed a quaternion-based procedure that, as our proposal, isolates the effect of magnetic disturbances over inclination (pitch and roll) estimations, but it is more sensible to the drift at gyroscope signals. With respect to the methods that address the problem of yaw estimation errors that are related to gyroscope bias, a model-based gyro bias estimation method is used in our solution. In this way, the state vector in the SFA is increased by adding the gyro bias, thus estimating the orientation and the gyro bias simultaneously. Where the SFA selected is concerned, this is a dual-extended Kalman filter with gyro bias estimation, where the orientation is directly represented by the Euler angles. Kalman filters (KFs) and extended Kalman filters (EKFs) are the preferred sensor fusion methods for estimating the attitude (orientation) using IMUs [14,15].

The rest of the paper is organized as follows. In Section 2, the ultrasound probe design will be exposed, and the main components of the hardware design will be described. As a main characteristic, it can be considered as a low cost and powerful embedded system. One key element of this hardware are the sensors, among which the IMU device has a central role. When estimating the orientation of the probe, it will be useful to describe the relationship between the fixed frame and the frame attached to the body (probe). In addition, the algorithm that is needed to estimate the hand orientation in the embedded system will be exposed. In Section 3, the main results of the computed orientation will be shown. In addition, the global simulating environment will be presented, and a case of study of the use of the probe in a real image based diagnostic task will be exposed. Finally, the discussion and the main conclusions of the work will be presented.

## 2. Materials and Methods

### 2.1. Components of the HUSP Probe

When a rehabilitation specialist performs an MSK evaluation via USI, the professional’s first step is to move the US probe/transducer to the MSK area of interest. At this time, he rotates, with three degrees of freedom, the US probe until the ideal orientation is reached to correctly visualize the target US image. By means of our solution, the student must imitate (mimic) this behavior. The design objective is to reproduce the translation and rotation movements of the hand of the real specialist as he handles the probe using hardware components with a high performance/cost ratio.

The HUSP unit (Figure 1) includes a sensor module and a data acquisition and fusion module which are integrated into a custom housing, replicating a real transducer. In the 3D design of the probe, an important effort was made to achieve a high degree of realism and to optimize the arrangement of both modules. The use of 3D additive printing techniques has been very useful at the prototype design process. The HUSP is powered from a 5 V source.

The sensor module consists of a commercial inertial measurement unit (IMU) MPU-9150 from InvenSense (InvenSense Co. San Jose, CA, USA) with mutually orthogonal tri-axial accelerometer, gyroscope and magnetometer, which is used to estimate the 3D orientation (roll, pitch and yaw angles) of the probe (Figure 2). In addition, the hardware components from an optical mouse are included to detect the two-dimensional translation movement of the probe back and forth, left and right, relative to the underlying surface. In this way, we will capture the typical rotation and translation movements that a rehabilitation specialist performs when positioning an US probe.

The data acquisition and fusion module are supported by iMX233-OLinuXino-NANO (Olimex, Ltd., Plovdiv, Bulgaria): a Linux computing board that has been designed to be embedded in custom devices, thanks to its dimensions, light weight and complete capabilities. The MPU-9150 will be connected via the I2C interface available on the Olinuxino. The IMU sensors are sampled at a 30 Hz rate, which is fixed according to the speed at which the probe is moved by the specialist, typically at a maximum linear velocity around 1 cm/s and a maximum angular velocity around 1 rad/s. Thus, this sampling rate is sufficient for fidelity and for adequate repeatability of the movement when carried out by a human being, ensuring real-time computational response.

From the MPU-9150 sensor, the angular rate, the proper linear acceleration and the earth magnetic field vector are collected in *X*, *Y*, *Z* axes. Integrating the angular rate obtained from the gyro outputs is not an appropriate method for calculating the orientation of the probe replica because the gyro bias error causes the orientation error to diverge. A sensor fusion algorithm is used to resolve the entire three-dimensional orientation of the probe, where data from gyros, magnetometers and accelerometers are integrated to correct the bias, drifts, noise and the magnetic field distortions that affects these measurements. In this way, in the following section, we will present the used algorithm which is based on a two-stage extended Kalman filter. Its implementation will perform the data fusion task for the data provided by the IMU device. The body orientation will be represented using Euler angles. Figure 2 shows the frame attached to the probe (the hand of the specialist) which has been considered to define the orientation.

Events associated with the movement of the mouse are directly processed in the desktop application (by the OS services). With the data related to the orientation and translation movements, the desktop application selects and displays to the user the corresponding US images, which have been pre-recorded with real US equipment and patients have been previously tagged with the position and orientation data.

### 2.2. Calculation of the Probe Orientation by IMU Sensor

The orientation of the probe can be defined by Euler (Tait–Bryan) angles: roll ϕ, pitch θ and yaw ψ that represent the relative attitude of a rotating frame *b*, which is attached at the moving body (probe), with respect to a fixed original frame *n* (usually named inertial). The transformation matrix $E_{n}^{b}$ from the fixed frame *n* to the body frame *b* is defined by three elemental rotations: first rotation $E_{n}^{2}$ about axis *z* by ψ, then a rotation $E_{2}^{3}$ around the new axis $y_{2}$ by θ and, finally, rotation $E_{3}^{b}$ around the new axis $x_{3}$ by ϕ:(1)$$E_{n}^{b}=E_{3}^{b}\left(\phi \right)E_{2}^{3}\left(\theta \right)E_{n}^{2}\left(\psi \right)=\;\left(\begin{array}{ccc} \cos\theta \cos\psi & \cos\theta \sin\psi & -\sin\theta \\ \sin\phi \sin\theta \cos\psi -\cos\phi \sin\psi & \sin\phi \sin\theta \sin\psi +\cos\phi \cos\theta & \sin\phi \cos\theta \\ \cos\phi \sin\theta \cos\psi +\sin\phi \sin\psi & \cos\phi \sin\theta \sin\psi -\sin\phi \cos\psi & \cos\phi \cos\theta \end{array}\right)$$
where
(2)$$E_{3}^{b}\left(\phi \right)=\left(\begin{array}{ccc} 1 & 0 & 0\\ 0 & \cos\phi & \sin\phi \\ 0 & -\sin\phi & \cos\phi \end{array}\right)$$
(3)$$E_{2}^{3}\left(\theta \right)=\left(\begin{array}{ccc} \cos\theta & 0 & -\sin\theta \\ 0 & 1 & 0\\ \sin\theta & 0 & \cos\theta \end{array}\right)$$
(4)$$E_{n}^{2}\left(\psi \right)=\left(\begin{array}{ccc} \cos\psi & \sin\psi & 0\\ -\sin\psi & \cos\psi & 0\\ 0 & 0 & 1 \end{array}\right)$$

Let ω with components $\left(\omega_{x},\omega_{y},\omega_{z}\right)$ be the rotation rate on the probe frame *b*. Let $\left(\dot{\phi },\dot{\theta },\dot{\psi }\right)$ be the derivatives of the Euler angles, where each component is the magnitude of the angular velocity in the inertial frame (in which the angle is defined). The on-board rotation rate [16] can be expressed in terms of the derivative of Euler angles as
(5)$$\left[\begin{array}{c} \omega_{x}\\ \omega_{y}\\ \omega_{z} \end{array}\right]=\left[\begin{array}{c} \dot{\phi }\\ 0\\ 0 \end{array}\right]+E_{3}^{b}\left(\phi \right)\left[\begin{array}{c} 0\\ \dot{\theta }\\ 0 \end{array}\right]+E_{3}^{b}\left(\phi \right)E_{2}^{3}\left(\theta \right)\left[\begin{array}{c} 0\\ 0\\ \dot{\psi } \end{array}\right]$$

By substituting Equations (Equation 2) and (Equation 3) into Equation (Equation 5), the relationship between rate change for Euler angles with the measured angular rates will be obtained as
(6)$$\left[\begin{array}{c} \dot{\phi }\\ \dot{\theta }\\ \dot{\psi } \end{array}\right]=\left[\begin{array}{ccc} 1 & \sin\phi \tan\theta & \cos\phi \tan\theta \\ 0 & \cos\phi & -\sin\phi \\ 0 & \sin\phi \sec\theta & \cos\phi \sec\theta \end{array}\right]\left[\begin{array}{c} \omega_{x}\\ \omega_{y}\\ \omega_{z} \end{array}\right]$$

Taking into account the initial conditions, Equation (Equation 6) can be integrated obtaining the dynamical evolution of the probe orientation (ϕ,θ,ψ) from gyro measurement $\left(\omega_{x},\omega_{y},\omega_{z}\right)$. As all real components in real scenarios, MEMS gyros show offsets, drifts and noise, which in numerical integration, cause tracking errors in the orientation which will increase with the time.

As a powerful solution, we use a well-known sensor data fusion algorithm, Extended Kalman Filtering (EKF), that will combine the data provided by accelerometers and magnetometers, which act as measurements of outputs, in a correction stage that will minimize the tracking error.

### 2.3. Kalman Filter in Tracking of Probe Orientation

To fuse the information of the three IMU sensors, we will use a dual-extended Kalman filter (Figure 3). The first filter, called Tilt EKF, will compute pitch (θ) and roll (ϕ) taking as input the gyro data at the prediction stage and it will use the accelerometer data at the correction phase. The second filter, named Heading EKF, will compute yaw angle (ψ) by integrating over the time the gyro information and the Tilt Filter output (θ,ϕ) at the prediction stage and it will use the magnetometer data at the correction phase. In this way, our method decouples the estimation of the pitch and roll angles from the magnetic disturbance, using the two-step orientation estimation.

#### 2.3.1. EKF Tilt

System state is defined [16,17,18] by using the third column of the director cosines matrix (Equation (Equation 1)), so
(7)$$x=\left[\begin{array}{c} x_{1}\\ x_{2}\\ x_{3} \end{array}\right]=\left[\begin{array}{c} -\sin\theta \\ \sin\phi \cos\theta \\ \cos\phi \cos\theta \end{array}\right]$$

If x is derived, then
(8)$$\dot{x}=\left[\begin{array}{c} -\dot{\theta }\cos\theta \\ \dot{\phi }\cos\phi \cos\theta -\dot{\theta }\sin\phi \sin\theta \\ -\dot{\phi }\sin\phi \cos\theta -\dot{\theta }\cos\phi \sin\theta \end{array}\right]$$

By substituting expressions $\dot{\theta }$ and $\dot{\phi }$ from Equation (Equation 6) into Equation (Equation 8), it will be obtained
$$\dot{x}=\left[\begin{array}{ccc} 0 & \omega_{z} & -\omega_{y}\\ -\omega_{z} & 0 & \omega_{x}\\ -\omega_{y} & \omega_{x} & 0 \end{array}\right]\left[\begin{array}{c} x_{1}\\ x_{2}\\ x_{3} \end{array}\right]$$

If a linearization procedure is carried out, the following expression will be obtained
(9)$$\Delta \dot{x}=\left[\begin{array}{ccc} 0 & \omega_{z} & -\omega_{y}\\ -\omega_{z} & 0 & \omega_{x}\\ -\omega_{y} & \omega_{x} & 0 \end{array}\right]\left[\begin{array}{c} \Delta x_{1}\\ \Delta x_{2}\\ \Delta x_{3} \end{array}\right]+\left[\begin{array}{ccc} 0 & -x_{3} & x_{2}\\ x_{3} & 0 & -x_{1}\\ x_{2} & -x_{1} & 0 \end{array}\right]\left[\begin{array}{c} \Delta \omega_{x}\\ \Delta \omega_{y}\\ \Delta \omega_{z} \end{array}\right]$$
which corresponds to the linearized system model
(10)$$\Delta \dot{x}=F\Delta x+B\Delta u$$
where Δu is the gyro information $\left(\Delta \omega_{x},\Delta \omega_{y},\Delta \omega_{z}\right)$, Δx is the state, and *F* and *B* the corresponding matrices appearing in Equation (Equation 9). The Δ symbol preceding the variables refers to deviation variables with respect to the linearization point (e.g., $\Delta u=u-\bar{u}$).

When discretizing the state-space model (Equation (Equation 10)), we take the solution x(t) of the state equation that uses the state-transition matrix $e^{Ft}$. In order to obtain the discrete model in a simple and computationally tractable way, the following approximation has been considered $e^{F\Delta T}\approx \left(I+F\Delta T\right)$. Hence, the approximate discrete model is
(11)$$x_{k+1}=\left(F_{k}\Delta T+I\right)x_{k}+B_{k}u_{k}\Delta T$$
where ΔT is the sampling time; $F_{k}$ and $B_{k}$ are the matrix at instant *k*
$$F_{k}=\left[\begin{array}{ccc} 0 & \omega_{z_{k}} & -\omega_{y_{k}}\\ -\omega_{z_{k}} & 0 & \omega_{x_{k}}\\ -\omega_{y_{k}} & \omega_{x_{k}} & 0 \end{array}\right]$$
$$B_{k}=\left[\begin{array}{ccc} 0 & -\cos\phi_{k}\cos\theta_{k} & \sin\phi_{k}\cos\theta_{k}\\ \cos\phi_{k}\cos\theta_{k} & 0 & \sin\theta_{k}\\ \sin\phi_{k}\cos\theta_{k} & \sin\theta_{k} & 0 \end{array}\right]$$

By defining a reduced system state, as shown in Equation (Equation 7), the system model, used in the EKF prediction phase, depends only on ϕ and θ and it is independent of ψ. The state at the instant k+1 depends on the state and the gyro data at the instant *k*.

At the EKF correction stage, the accelerometer data will be used to compute estimations of ϕ and θ. As accelerometer measurements are made respect to the body frame, we must transform the gravity vector from the fixed frame *n* to the body frame *b*. To do that, the following expression can be used:(12)$$\left[\begin{array}{c} A_{x}\\ A_{y}\\ A_{z} \end{array}\right]=E_{n}^{b}\left[\begin{array}{c} 0\\ 0\\ -g \end{array}\right]=-g\left[\begin{array}{c} -\sin\theta \\ \sin\phi \cos\theta \\ \cos\phi \cos\theta \end{array}\right]=-g\left[\begin{array}{c} x_{1}\\ x_{2}\\ x_{3} \end{array}\right]$$
where $E_{n}^{b}$ represents the matrix in Equation (Equation 1) and *g* will designate the gravity value. Given the state definition that appears in Equation (Equation 7) at the instant *k*, the observation model will be given by
(13)$$z_{k}=H_{k}x_{k}$$
where $z_{k}$ represents the accelerometer measure ${\left[A_{x},A_{y},A_{z}\right]}^{T}$ at the instant *k* and the observation matrix will be $H_{k}=-g$.

#### 2.3.2. EKF Heading

Since the gyro has offset errors (bias),
(14)$${\dot{\psi }}^{s}=\dot{\psi }-{\dot{\psi }}^{e}$$
can be written, where ${\dot{\psi }}^{e}$ and ${\dot{\psi }}^{s}$ are, respectively, the offset error and the estimated value of $\dot{\psi }$. Once the devices have reached the right temperature, we have found that assuming a time-invariant bias is valid.

Considering yaw estimation errors related to gyroscope bias, we define the discrete system state as
(15)$$x_{k}=\left[\begin{array}{c} \psi_{k}^{s}\\ {\dot{\psi }}_{k}^{e} \end{array}\right]$$
where $\psi_{k}^{s}$ is the estimated yaw angle at instant *k* and ${\dot{\psi }}_{k}^{e}$ is the offset error of $\dot{\psi }$ at instant *k*. By sustituting the discrete derivative of ${\dot{\psi }}_{k+1}^{s}$ at instant k+1
(16)$${\dot{\psi }}_{k+1}^{s}=\frac{\psi_{k+1}^{s}-\psi_{k}^{s}}{\Delta T}$$
in Equation (Equation 14), one gets
(17)$$\psi_{k+1}^{s}=\left({\dot{\psi }}_{k+1}-{\dot{\psi }}_{k+1}^{e}\right)\Delta T+\psi_{k}^{s}$$

If we express the Equation (Equation 17) in matrix form, assuming a time-invariant bias ${\dot{\psi }}_{k+1}^{e}={\dot{\psi }}_{k}^{e}$
(18)$$\left[\begin{array}{c} \psi_{k+1}^{s}\\ {\dot{\psi }}_{k+1}^{e} \end{array}\right]=\left[\begin{array}{cc} 1 & -\Delta T\\ 0 & 1 \end{array}\right]\left[\begin{array}{c} \psi_{k}^{s}\\ {\dot{\psi }}_{k}^{e} \end{array}\right]+\left[\begin{array}{c} \Delta T\\ 0 \end{array}\right]{\dot{\psi }}_{k+1}$$
we obtain the system model that is used in the prediction stage of EKF Heading
(19)$$x_{k+1}=F_{k}x_{k}+B_{k}u_{k}$$
where $x_{k}$ is defined in Equation (Equation 15) and the input $u_{k}$ corresponds to ${\dot{\psi }}_{k+1}$. This input can be computed by substituting the third row from Equation (Equation 6) and by using of state vector x defined at EKF Tilt in Equation (Equation 7), as
(20)$$\dot{\psi }=\frac{x_{2}}{x_{2}^{2}+x_{3}^{2}}\omega_{y}+\frac{x_{3}}{x_{2}^{2}+x_{3}^{2}}\omega_{z}$$

Input $\dot{\psi }$ depends on the components $\left(\omega_{y},\omega_{z}\right)$ of angular rate of the gyro and, additionally, it depends on roll (ϕ) and pitch (θ), through $\left(x_{2},x_{3}\right)$, which are calculated by the EKF Tilt. Therefore, EKF Heading must be calculated after the evaluation of the EKF Tilt.

At the observation stage, the information provided by the magnetometer will be used to compute an estimation for ψ. Again, it will be necessary to make a transformation for the vector that points to the magnetic north $H_{n}={\left[H_{N}H_{E}H_{D}\right]}^{T}$, taken at the fixed frame, with respect to body frame $H_{b}$ by means of
(21)$$H_{b}=E_{n}^{b}H_{n}$$
where $E_{n}^{b}$ is the matrix of Equation (Equation 1) and so the resulting matrix is [17,18]:(22)$$H_{b}=\left[\begin{array}{c} H_{N}\cos\theta \cos\psi +H_{E}\cos\theta \sin\psi -H_{D}\sin\theta \\ H_{N}\left(\sin\phi \sin\theta \cos\psi -\cos\phi \sin\psi \right)+H_{E}\left(\sin\phi \sin\theta \sin\psi +\cos\phi \cos\psi \right)+H_{D}\sin\phi \cos\theta \\ H_{N}\left(\cos\phi \sin\theta \cos\psi +\sin\phi \sin\psi \right)+H_{E}\left(\cos\phi \sin\theta \sin\psi -\sin\phi \cos\psi \right)+H_{D}\cos\phi \cos\theta \end{array}\right]$$

This model is nonlinear, so it is necessary to linearize it, using
(23)$$H_{b}=f\left(\psi^{s},{\dot{\psi }}^{e}\right)\Rightarrow \nabla H_{b}={\left[\frac{\partial H_{b}}{\partial \psi^{s}},\frac{\partial H_{b}}{\partial {\dot{\psi }}^{e}}\right]}_{\left({\bar{\psi }}^{s},{\bar{\dot{\psi }}}^{e}\right)}$$
where
(24)$$\nabla H_{b}={\left[\begin{array}{cc} -H_{N}\cos\theta \sin\psi +H_{E}\cos\theta \cos\psi & 0\\ H_{N}\left(-\sin\phi \sin\theta \sin\psi -\cos\phi \cos\psi \right)+H_{E}\left(\sin\phi \sin\theta \cos\psi -\cos\phi \sin\psi \right) & 0\\ H_{N}\left(-\cos\phi \sin\theta \sin\psi +\sin\phi \cos\psi \right)+H_{E}\left(\cos\phi \sin\theta \cos\psi +\sin\phi \sin\psi \right) & 0 \end{array}\right]}_{\left({\bar{\psi }}^{s},{\bar{\dot{\psi }}}^{e}\right)}$$

Hence, the observation matrix at the EKF Heading is as shown in Equation (Equation 24).

## 3. Results

### 3.1. Real-Time Orientation Estimation

This section focuses on the experimental validation of the proposed algorithm to calculate the complete probe orientation by the sensor fusion of data provided by IMU. In order to perform the tests, we have built an experimental platform (Figure 4) that allows the probe to rotate, in a controlled way, with different orientations. The designed HUSP (d) was placed on a support piece (c) that is coupled to a servomotor (b). The support is made of plastic and has been designed for the servomotor and the probe was separated at a certain distance (about 25 cm) in order to ensure that the electromagnetic field generated by the motor operation does not interfere with the measure of the earth magnetic field. A Rasberry Pi (a) serves as motor supervisor, so the HUSP rotates solidarity. With this experimental development, we can control the rotation angle of the probe and the time it takes to get its final orientation at the set speed. A script for the servomotor to rotate a certain angle at a certain speed and stop a certain time was programmed in the Rasberry Pi. It was verified that the values fixed in the script were met correctly by the servomotor, as it is obvious.

A test procedure was designed that consisted of rotating the probe on each of the three axes individually, as shown in Figure 2, with different angle values and with the consideration of different speeds. Before starting the tests, the probe was fixed to the platform to get the three different orientations by means of different support pieces. In Figure 4c, the support piece for yaw test is shown.

The first test consisted of placing the probe vertically, modifying the yaw angle (relative to Earth’s magnetic field): the probe has an initial orientation $\left[\phi ,\theta ,\psi \right]=\left[0,0,75^{\circ }\right]$; then, it rotates with ω=0.3 rad/s, until it reaches the orientation $\left[\phi ,\theta ,\psi \right]=\left[0,0,34^{\circ }\right]$ and it stops for 10 s; afterwards, the probe returns to its initial orientation at the same speed, it stops for 10 s; then, it rotates with the same speed as far as $\left[\phi ,\theta ,\psi \right]=\left[0,0,-8^{\circ }\right]$, it stops for 10 s and, again, it returns to its initial orientation. In Figure 5, measurements ($A_{x},A_{y},A_{z}$) vs. time for accelerometer (Figure 5a in m/s${}^{2}$ vs. s), gyro ($\omega_{x},\omega_{y},\omega_{z}$) (Figure 5b in rad/s vs. s) and magnetometer ($M_{x},M_{y},M_{z}$) (Figure 5c in μT vs. s) can be observed, where an important presence of white noise can be seen. After the application of the EKF (that was described in the previous section), the set of values that represent the orientation of the probe replica is calculated. In Figure 5d, the time evolution of the orientation calculation with EKF is shown, where ψ varies from $75^{\circ }$ to $34^{\circ }$, then from $75^{\circ }$ to $-8^{\circ }$, stopping for 10 s at each change of orientation.

In the second test, the probe was arranged horizontally, by means of other support pieces, in order to check the roll angle and the pitch angle. First, the *x*-axis of the body frame was made to coincide with the axis of rotation of the servomotor (roll angle variations only) and, later, the *y*-axis (pitch angle variations only). In this second test, first, the probe rotates from [ϕ,θ,ψ]=[0,0,0] to $\left[-40^{\circ },0,0\right]$ with ω=0.3 rad/s; then, it stops and it returns back to initial orientation; afterwards, it turns to $\left[-80^{\circ },0,0\right]$, it stops and goes back to initial orientation. In Figure 6, time evolution of accelerometer measurements (Figure 6a in m/s${}^{2}$ vs. s), gyro (Figure 6b in rad/s vs. s) and magnetometer (Figure 6c in μT vs. s) can be observed, with an important presence of white noise. In Figure 6d, the orientation calculation with EKF is displayed, where ϕ varies from $0^{\circ }$ to $-40^{\circ }$, then from $-80^{\circ }$ to $0^{\circ }$, stopping for 10 s at each change of orientation.

In this second test, later, only variations in the pitch angle have been made: $\theta =40^{\circ }$ and $\theta =80^{\circ }$ turns with ω=0.6 rad/s. In Figure 7, the IMU measurements (Figure 7a for accelerometer, Figure 7b for gyro, Figure 7c for magnetometer) and calculated orientations (Figure 7d) are shown.

For the accelerometer, gyroscope and magnetometer measurements, which correspond to the three tests (Figure 5, Figure 6 and Figure 7a–c), it is possible to observe the significant presence of white noise. Once the proposed SFA has been applied (Figure 5, Figure 6 and Figure 7d), note that the noise has practically disappeared. In order to get this result by means of the dual EKF, it is very important to compute the correct covariance matrices. The variance of each input was calculated with the HUSP at a static position (that is, we assume a white noise behaviour). We have programmed the servomotor with different speeds to check that the angular velocity and the time between different orientation changes are correct (that is, artificial delays do not exist).

### 3.2. The Complete System Evaluation

The HUSP device has been integrated into the complete platform to instruct future rehabilitation specialists in MSK ultrasound imaging. In Figure 8, one can see a scheme of our US training tool. The computer embedded in the probe replica, with the IMU data, calculates the probe attitude and sends it to the desktop computer through UART-USB communication. The probe replica incorporates the board of an optical mouse, connected by USB to the desktop computer, to obtain the relative position of the probe replica in the model.

Using the position and orientation of the physician trainee’s movements, the US images corresponding to that position and orientation are displayed on the training environment screen. Previously, medical specialists at USI have recorded images of a person, using real ultrasound equipment and these images have been tagged with the position and orientation where they were obtained.

The training platform developed includes a set of theoretical contents about anatomy necessary to practice the recognition of MSK structures through USI. To facilitate the learning process, a 3D model of the human body with neuromuscular systems (NMS) has also been included. A team of rehabilitation specialists, radiologists, traumatologists and anesthesiologists have designed a set of practical cases in which students will learn interactively how to position and orientate the US probe to obtain the ideal USI that will allow them to identify the anatomy of a muscle section (or other items such as nerves or blood vessel) they are visualizing. In the first stage, practical cases have focused principally on different NMS of the upper and lower limbs. This fact must not be seen as a restriction since these body parts provide a sufficiently complex scenario.

As the student moves and orients HUSP over the mouse pad, he receives visual feedback (mainly composed by US images) in the graphical interface of the training application. Orientation and position tracking of the probe make it possible to show, on the desktop screen (Figure 8), a virtual ultrasound probe on the upper and lower limbs of a human model. In this figure, a muscle is identified, but nerves and veins or arteries are also identified in the training platform when they are distinguishable. In addition, information to make an interpretation of the ultrasound images corresponding to the human body of the case study is also displayed.

As an aid to learning, a set of gadget tools have been included to facilitate the location and identification of MSK structures. When probe location is considered, at the moment that the trainee places and orients the probe in the region of interest, a green box is shown on the ultrasound image; red and yellow indicate whether it is distant or close, to fix the attention of the student. In addition, it is possible to show the user the path to be taken for the MSK localization corresponding to that practical case. As far as identification is concerned, the contours of the MSK structures are superimposed on the ultrasound image. In this way, the training platform makes it possible for students to create a virtual haptic sensation with which the physician can enhance his mental model at USI work.

## 4. Discussion

Diagnosis and treatment of musculoskeletal pathologies, in medical rehabilitation tasks, have benefited from the use of image treatment techniques, where USI stands out for being widely accepted by specialists and patients since it is not a technique based on ionizing radiation. This technology adds the needed accuracy on diagnosis and certainty to guide the professional at therapeutics interventional procedures. However, this technique has a steep learning curve, since the specialists must acquire knowledge that allows them to relate the anatomy of the human body (that is, a three-dimensional space) with the one they will see in the two-dimensional ultrasound images. In order to achieve this mental model, that relates the hand movements and probe orientation with the desired US image, our tool provides an economic learning path with good results.

The learning process can occur at any place with lower hardware requirements than other proposals such as [19], without loss of precision in the capture of movements. Data on trainee’s hand movements related to 3D orientation (roll, pitch and yaw angles) are captured by means of an inertial sensor based on microelectromechanical systems (MEMS), which meets the cost, payload and space restrictions imposed on HUSP. In [20], several devices are described in such a way that our base device (MPU-9150) is included. An onboard motion fusion module, named the digital motion processor (DMP), and calibration firmware was integrated into MPU-9150, which enables users to quickly develop motion-based functionality. They show that the performance of the set constituted by this MEMS and its DMP is very poor, but it has interesting advantages such as price, availability or dimensions that we have exploited.

Thus, we have needed to develop our own sensor fusion algorithm which, as shown, has good performance. This is due to the solid work done with our algorithmic proposal: A dual EKF that decouples the estimation of the pitch and roll angles from the magnetic disturbance, using the two-step orientation estimation. In the first step, the EKF estimates pitch and roll using gyro data at the prediction stage and accelerometer data at the correction phase. In the second step, the EKF estimates yaw using, at the prediction stage, the gyro data and the estimated pitch and roll and, at the correction stage, the magnetometer data. The definition of a reduced system state (as in [16,17,18]) at the first EKF achieves: (1) The system model depends only on the pitch and roll angle and it is independent of the yaw angle; (2) the system model is easily linearized; and (3), in the measurement model, the accelerometer output is used directly without further complex transformations. Unlike [17], at the second EKF, the state of the system includes gyro bias to solve the problem of estimation errors related to gyro bias. Along with the sound formalization of the proposed SFA, an exhaustive calibration of the mathematical models has been carried out.

With all of them, the trainee sensation is completely free of delays as shown in (http://gro.usal.es/videos/probe.mp4). By considering the complete software interface that has been developed (with an impressive HMI), our proposal system can be used, at any time, by specialists that want to acquire the mental model needed by USI based rehabilitation tasks.

## 5. Conclusions

This paper describes the development of a haptic US probe integrated into a complete platform and dedicated to training in musculoskeletal ultrasonography for rehabilitation. The device captures rotation and translation movements that a physician performs when he moves the probe in the context of diagnostic work. A MEMS-based IMU sensor together with a robust SFA (that reduces the magnetic disturbance and bias errors influence) that is executed at an advanced embedded system makes it possible. A great effort has been done to get that the HUSP aspect to act and feel like a real probe, not only in the shape but also in a realistic operation. The performance/cost rate has been highly optimized, so that it will be available for many people. Our solution provides autonomous learning anytime, anywhere, so the trainee may acquire the needed capabilities on US diagnostic techniques in a fast and complete way. These techniques are very useful for efficient early treatment in a large number of medical situations.

## 6. Patents

The HUSP device is protected by the utility model number 201700521 granted by the OEPM (Oficina Española de Marcas y Patentes-Spanish Trademark and Patent Office) on 23 April 2018.

## Figures and Tables

**Figure 1 sensors-19-00101-f001:**
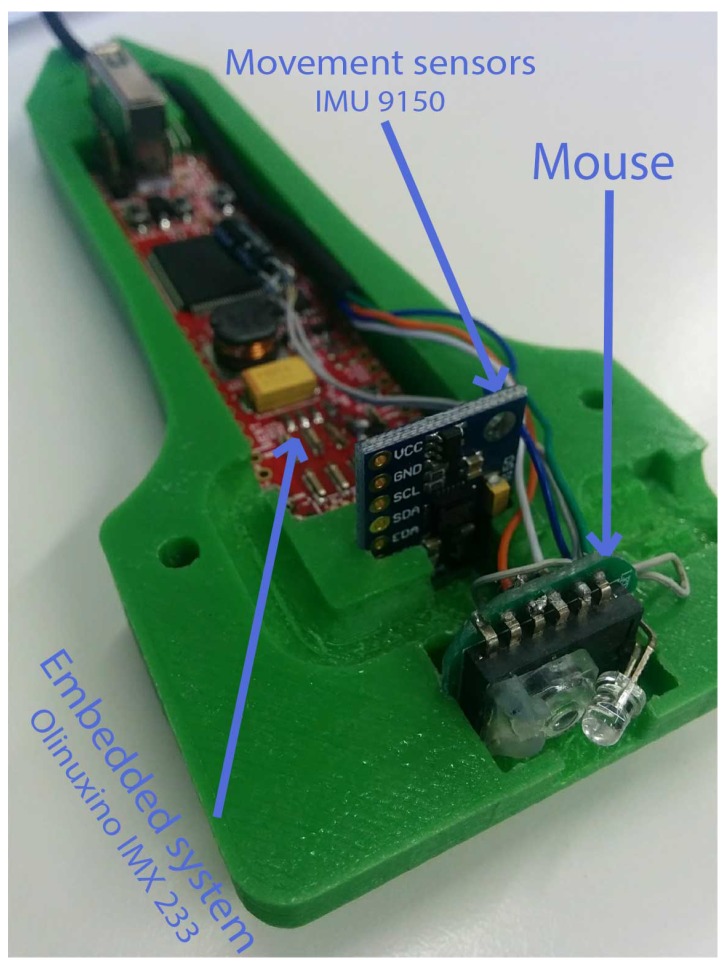
Arrangement of the hardware modules inside HUSP: optical mouse board, IMU and Linux embedded computer.

**Figure 2 sensors-19-00101-f002:**
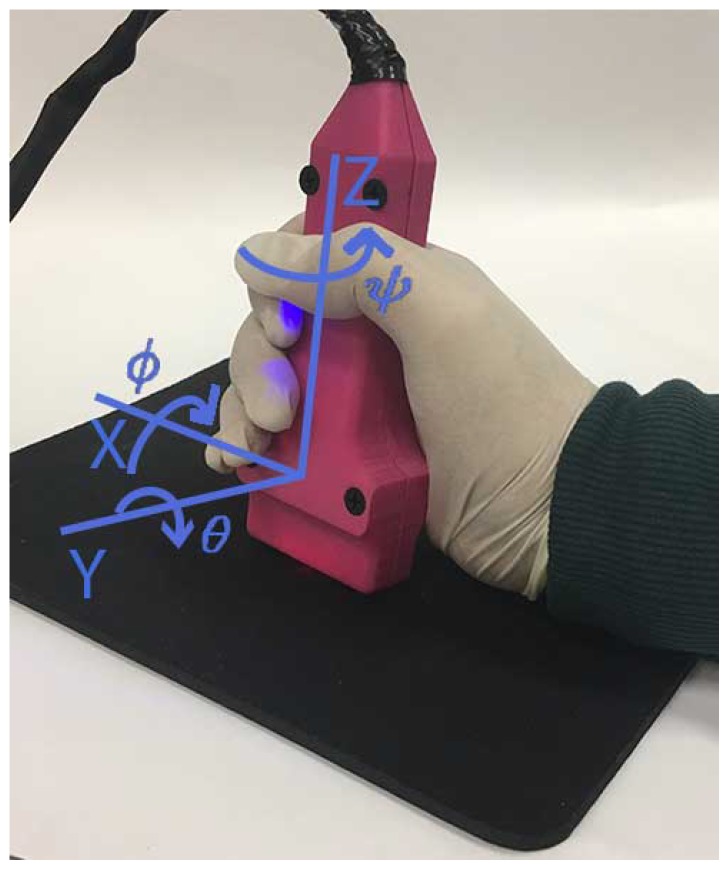
Rotation movements of the probe in the hand of a specialist and body frame.

**Figure 3 sensors-19-00101-f003:**
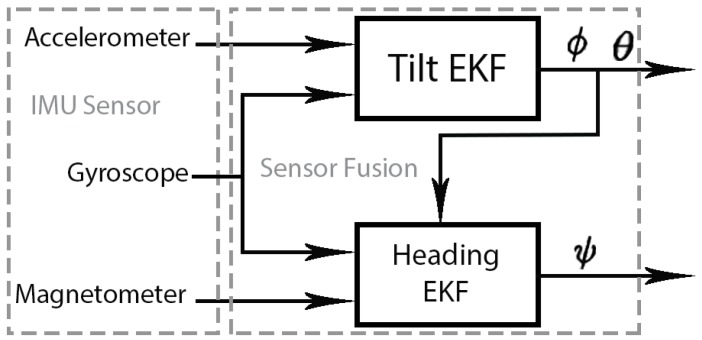
Scheme of EKF used in the orientation estimation.

**Figure 4 sensors-19-00101-f004:**
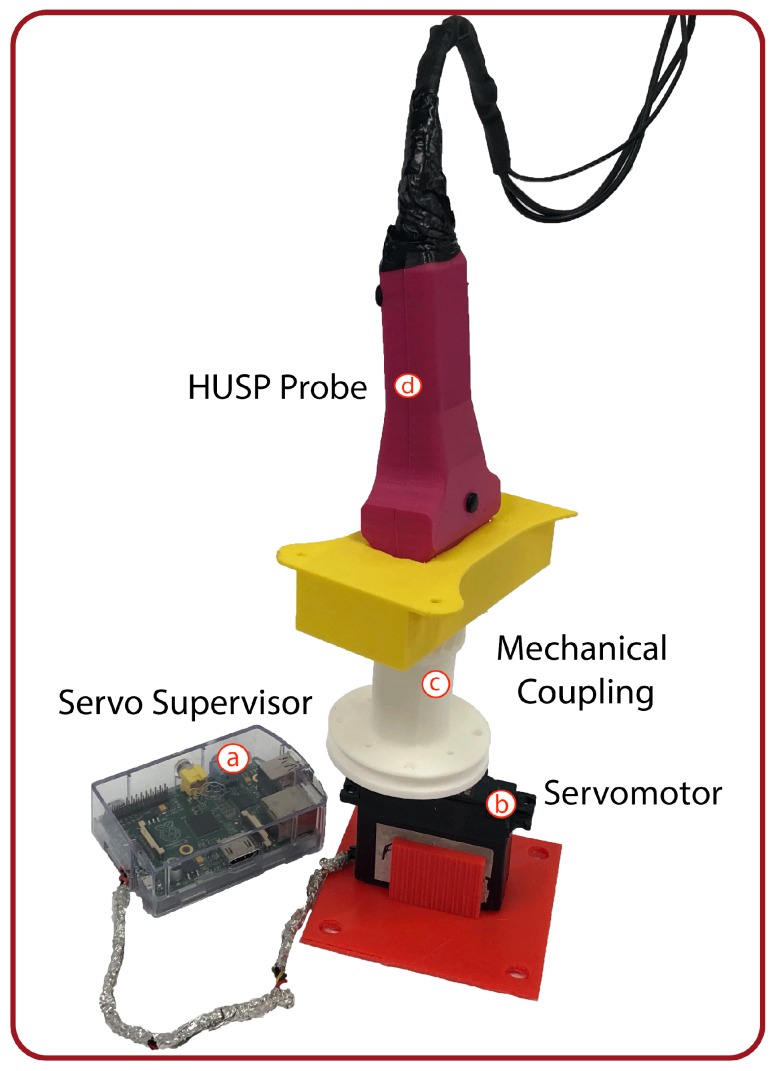
Experimental platform built to validate the sensor fusion algorithm.

**Figure 5 sensors-19-00101-f005:**
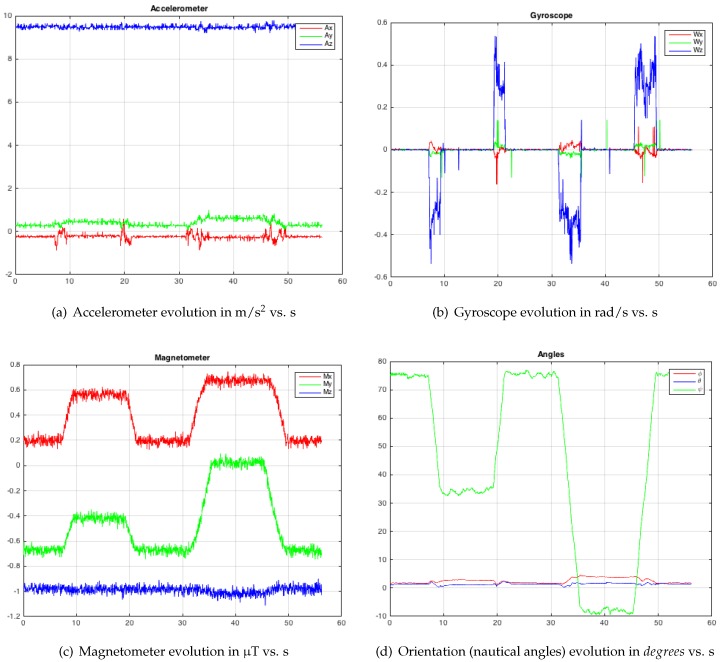
Test 1—Modifying the yaw angle (ψ turn): data collected with the IMU and the orientation estimation.

**Figure 6 sensors-19-00101-f006:**
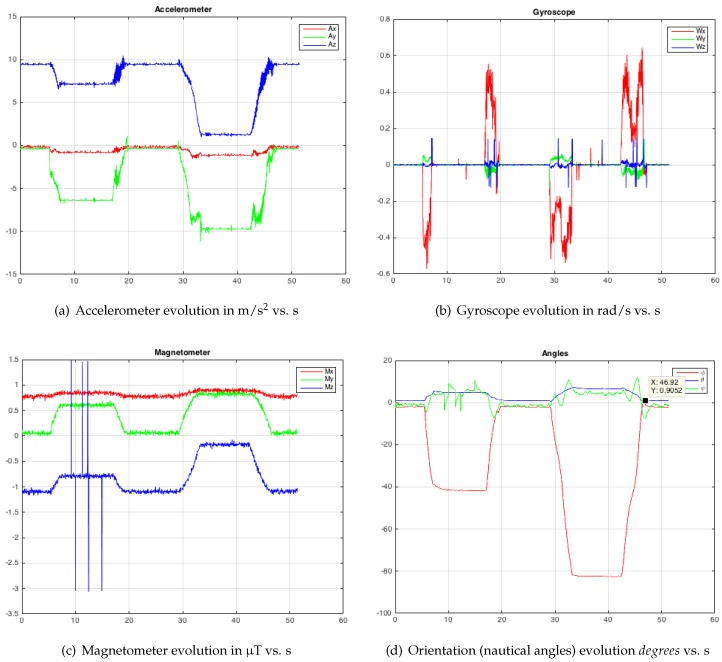
Test 2—Modifying the roll angle (ϕ turn).

**Figure 7 sensors-19-00101-f007:**
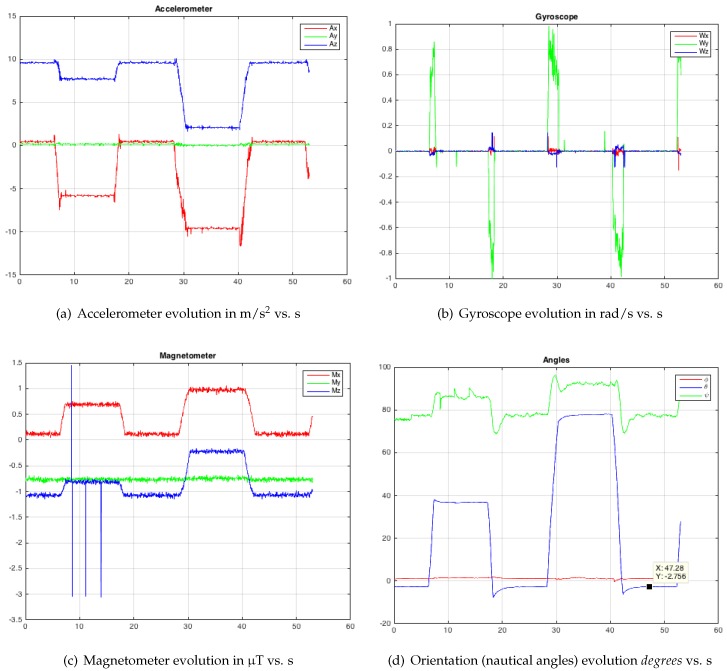
Test 2—Modifying the pitch angle (θ turn).

**Figure 8 sensors-19-00101-f008:**
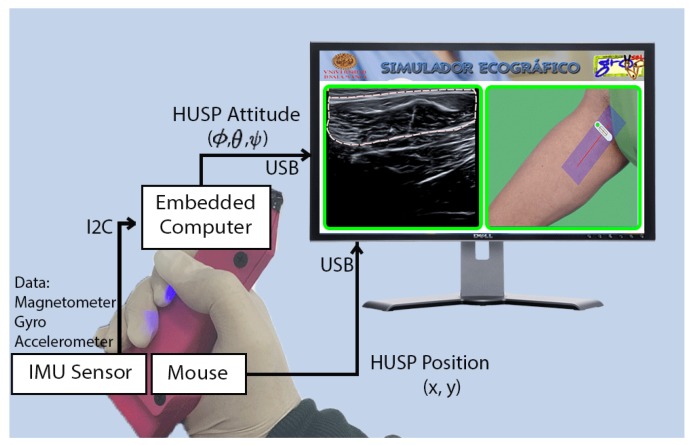
Global scheme of the US probe-based training platform.

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
