# Peer review of "HUSP: A Smart Haptic Probe for Reliable Training in Musculoskeletal Evaluation Using Motion Sensors"

_sensors, 2018, doi:10.3390/s19010101_

Round 1
Reviewer 1 Report
The current manuscript presents a very interesting application of inertial sensor fusion. The value of the ultrasound training system and the need for orientation estimation are well explained. The sensor fusion algorithm is presented in detail. However, its novelty is not explained and experimental results are not analyzed. Instead, the introduction, discussion and conclusion focus entirely on the value of the ultrasound training system and barely on the sensors-related aspects of the work. Therefore, my main suggestion is that the manuscript should be revised to strengthen the sensors-related contribution. In the following, I provide more detailed comments:
1-7: The introductory part of the abstract is unusually long. Please make it more concise.
14: Please be more precise what "complete three-dimensional attitude" means. In most recent papers attitude would be only the two-dimensional information of the inclination (roll and pitch in ypr angles) of the object, and the third piece of information woult be called heading (yaw in ypr angles). Please state very briefly in the abstract whether you determine the heading and whether you use magnetometers (and if yes, which assumptions you make regarding the local magnetic field).
71: The IMU is based on MEMS, not vice versa.
The introduction explains why the US training system is developed and provides nine references in this context. It might be acceptable if the paper was submitted to an US journal. However, it was submitted to a sensors journal, and according to the abstract a main contribution is the sensor fusion algorithm, which is also the focus of the methods section. Therefore, the reader expects a firm review of the current state of the art in inertial sensor fusion including recent advancements in IMU orientation estimation, i.e. at least DOIs 10.1109/ES.2017.37 10.1016/j.ifacol.2017.08.1534 10.3390/s18010076 10.1561/2000000094. It should then be explained how the proposed algorithm relates to these recent works and to references 10–12.
109: Why is the sampling rate as low as 30 Hz? Using a larger sampling rate should be advantageous.
148: The delta variables should be introduced properly.
The discretization of linear differential equations can be done in many different ways. Which method did you employ? How are F_k and B_k calculated? Explain I even if it seems obvious.
154: Do you assume time-invariant bias? Please briefly motivate the model equations.
169: What precisely is "a certain distance"?
Please add text labels to Figure 4 explaining the components of the depicted objects.
178: "dps" is not a standard unit.
179: How did you determine the ground truth orientation angles? Describe this step in the manuscript, it is important.
Figures 4, 5, and 6 present results of a validation experiment. However, the results are not interpreted or analyzed. Are all results reasonable? How accurate is the system? In Section 3, not only the value of the training system should be discussed, but also the quality of the technical solutions and results.
Language should be carefully checked and revised. Here are some examples from the first pages:
18: "lets"
21: "an episode"
26: "our functionality"
34: "The" missing.
36: "no[t] radiation"
defiined, connnected, calcula pitch y roll realizando, it will obtained, followin, as appear at, made respect to, the third file, ...
Author Response
It is included at the attached file

Reviewer 2 Report
In this manuscript, the authors have designed a haptic probe to for applications in musculoeskeletal evaluataion using ultrasound imaging. Although, it is interesting and stimulating design also openning the way for future implications, I have some several major concerns to be taking into account in the study:
- There is no simulation on the tissue, how is the response of the probe?, The design is based on the components, there is no optimal design procedure, or finite element procedure to understand how is the response on tissue or even a different set of images analyzing the response in terms of displacements, or strain and so on.
-The components of the HUSP probe are detailed in a figure and in the text but a flow chart would facilitate the scheme of the design. 

-Also at least a statistical analysis on phatoms or in a mĂnimum simple is needed.
- the discussion needs more references in the field and consclusion where is the approach?
-In the conclusion where is the approach? And in the next paragraph is imprecise the way wich medical situations and why the technique is faster,
Main advantage of our solution lets to an autonomous learning at any place and moment, so the the trainee may acquire the needed capabilities on US diagnostic techniques in a faster and complete way. It must be considered that these techniques are very useful for efficient early treatment at a large medical situations. 

-The contribution and acknowledgement are empty
Author Response
It is included at the attached file

Round 2
Reviewer 1 Report
All questions and concerns have been addressed properly, and the quality of the manuscript has improved a lot. A few language inaccuracies (e.g. line 301) are left, the and the presentation of the results might be improved (e.g. the axis labels in Fig. 6 are not readable). I recommend the manuscript for publication after minor revisions.
Author Response
All the changes have been taken into account.

Reviewer 2 Report
The manuscript is better explained. However, I have some minor points to improve its quality:
-Acronyms should be defined if they are used in the abstract
-The introduction not provides enough references
-The size of equation 1 and 24should be rescaled
-In figures 5 and 6 the variables are not explained in the text, they have not enough size and the axes should be defined.
-243 I don't understand why different neuromuscular systems (NMS) of the upper and lower limbs -are the first step and then figure 8 just explores muscular aspects.
-294 What does mental model mean?
The section of patens, acknowledgement, author contribution, conflicts of interest and funders in lines 307-330 is empty, the template is added. It should be changed.
Author Response

(The authors gave the same response as above.)
